# Secure Voting Website Using Ethereum and Smart Contracts

**Abhay Singh** [1], **Ankush Ganesh** [1], **Rutuja Rajendra Patil** [2], **Sumit Kumar** [1,*], **Ruchi Rani** [3,4]
**and Sanjeev Kumar Pippal** [5]

[1] Symbiosis Institute of Technology, Pune, Symbiosis International (Deemed University),
Pune 412115, Maharashtra, India; abhaysengar3250@gmail.com (A.S.); masterankush0115@gmail.com (A.G.)

[2] Computer Science and Engineering-Artificial Intelligence and Machine Learning Department,
Vishwakarma Institute of Information Technology, Pune 411037, Maharashtra, India; rutujapat@gmail.com

[3] Department of Computer Science Engineering and Technology, School of Computer Science and Engineering,
Dr. Vishwanath Karad MIT World Peace University (MIT-WPU), Pune 411038, Maharashtra, India;
ruchiasija20@gmail.com

[4] Department of Computer Science, Indian Institute of Information Technology, Kottayam 686635, Kerala, India

[5] Department of Technology, NSBT, MGM University, Aurangabad 431005, Maharashtra, India;
sanpippalin@gmail.com

* Correspondence: er.sumitkumar21@gmail.com

**Abstract:** Voting is a democratic process that allows individuals to choose their leaders and voice their opinions. However, the current situation with physical voting involves long queues, paper-based ballots, and security challenges. Blockchain-based voting models have appeared as a method to address the limitations of traditional voting methods. As blockchain is distributed and decentralized, which uses hash functions for securing transactions, it dramatically improves the existing voting system. These digital platforms eliminate the need for physical presence, reduce paperwork, and ensure the integrity of votes through transparent and tamper-proof blockchain technology. This paper introduces a blockchain-based voting model to enhance accessibility, security, and efficiency in the voting process. The research focuses on developing a robust and user-friendly voting system by leveraging the advantages of decentralized technology. The proposed model employs Ethereum as the underlying blockchain platform through an innovative and iterative approach. The model uses Smart contracts to record and validate votes, while AI-based facial recognition technology is integrated to verify the identity of voters. Rigorous testing and analysis are conducted to validate the effectiveness and reliability of the proposed blockchain-based voting model. The system underwent extensive simulation scenarios and stress tests to evaluate its performance, security, and usability.

**Keywords:** e-voting; blockchain; EVM; digital platform; decentralized





## 1. Introduction

Elections are crucial, but a sizable portion of the population has little faith in the system, a grave issue for a nation. Even the most significant countries, such as India, are affected by a flawed electoral system. The fundamental problems with the current vote-casting device are vote falsification, election control, and poll space shooting. In the modern era, blockchain technology has been a revolutionary force dramatically impacting numerous industries, from banking to healthcare [1]. One of the most intriguing applications of blockchain technology has been its use in the voting system, allowing for voter authentication, improved security, and increased transparency [2,3]. Blockchain is a distributed ledger system that records and stores data securely and reliably. It is a decentralized system, meaning the data is stored on many computers rather than a single, central computer, making it virtually impossible to tamper with or manipulate. A blockchain voting system relies on this technology to provide secure, reliable voting [4]. Blockchain technology in the poll has been gaining attraction due to its promise of improved security, greater transparency, and more efficient management. Blockchain voting systems are more

secure than traditional voting systems because they provide a tamper-proof record of each vote stored securely on the blockchain and can be verified by anyone. This eliminates the possibility of faked or manipulated votes, as each ballot is transparently recorded on the blockchain for anyone to view.

Another advantage of blockchain voting systems is that they are much more efficient than traditional ones. They are faster, as votes can be recorded and stored securely on the blockchain in minutes or seconds, compared to conventional voting systems, which sometimes take days or weeks to be tallied. This makes it easier for officials to manage and monitor elections, as the votes are tabulated in real-time. In addition, blockchain voting systems can also be used to provide voter authentication. This ensures that only authorized voters can vote and are securely stored and validated. This eliminates the possibility of voter fraud, as votes can be tracked and verified to ensure that they are authentic [5–11].

Finally, blockchain voting systems provide greater transparency than traditional voting systems due to the distributed nature of the blockchain. All votes are securely stored on the blockchain, allowing anyone to view and verify them. This eliminates the possibility of rigged or manipulated polls, as anyone can view them and verify that they are accurate and authentic. In conclusion, blockchain voting systems offer numerous advantages over traditional voting systems, such as improved security, transparency, and efficiency. As technology evolves and improves, blockchain voting systems become increasingly popular and could soon become the norm in elections worldwide [12–14].

Section 1 represents a brief introduction, and Section 2 describes Blockchain in detail and its subsequent necessity. Section 3 iterates through the short report of the literature review. Section 4 outlines the proposed methodology and design. It also gives details of multiple phases and the level of work done in the technical aspect of the proposed system. Section 5 is an analysis and discussion of the theoretical and practical implementation of the proposed method; It also has a validation subsection. Finally, Section 6 is the conclusion, briefly summarizing the proposed model.

## 2. Blockchain

Blockchain can assist in implementing an immutable, transparent, and efficient system that cannot be hacked. Because blocks cannot be changed or deleted, blockchain can solve the problems in the traditional voting system. Blockchain is a distributed network composed of many interconnected nodes. A centralized approach does not control the network. If most nodes agree, the transaction allows users to be anonymous. Whenever there is a need for transparency and decentralized authentication and identification of users, we use blockchain. The transactions in a blockchain network are encrypted. In simpler words, blockchain is a shared database [15,16].

Core components of Blockchain Architecture:

- Node: Each user in a blockchain network is a node. A copy of the distributed ledger is shared with all the blockchain network nodes.
- Transaction: is the foundation of blockchain. In a blockchain, transaction details (asset, price, and ownership) are recorded and verified across all nodes.
- Block: numerous blocks in the blockchain network store information such as the hash of that block. Figure 1 represents blocks committed in a ledger with their hash Id.
- Chain: blocks in a specific order. Blockchain ensures these blocks' order by storing the previous node's hash in the current node.
- Miners: nodes that perform complex blockchain operations are responsible for verifying whether a transaction is valid.
- Consensus: Blockchain follows some algorithms to reach an agreement among nodes participating in a transaction. It's proof of work.

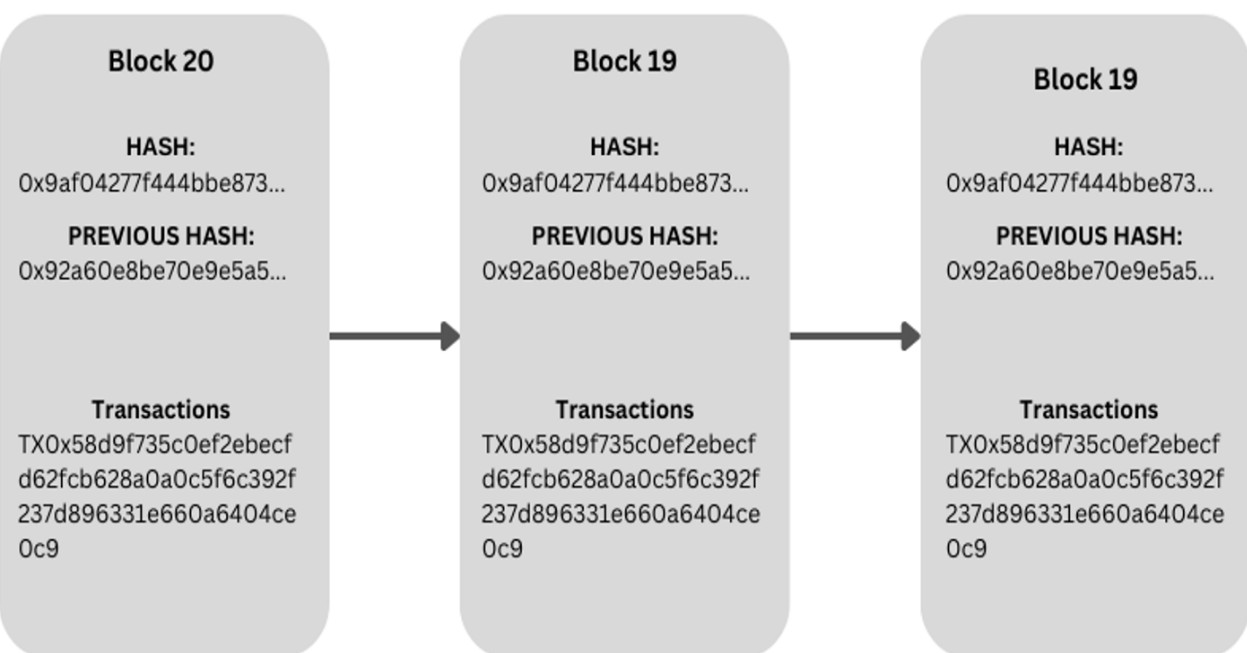

**Figure 1.** Blocks committed in a ledger.

Some common terminologies in a blockchain are:

1.  Smart Contract: Smart contracts are a simple logical bit of programs stored on each block of blockchain that run when some conditions are proper. They follow "if/when . . . then . . . " statements written on the blockchain and help automate the verification of parties' agreement. Automated execution of an agreement is done with the help of smart contracts; it ensures that each participant can be sure of the transaction's outcome. The irreversibility and traceability of these transactions increase the trust of the nodes in the network. Some advantages of smart contracts are making faster and better decisions, saving time, and lowering cost and risk.
2.  Wallet: Wallet is a collection of user identities. It allows users to store and manage their Bitcoin, Ether, and other cryptocurrencies.
3.  Truffle: It's a blockchain development environment. It also provides a testing framework and asset pipeline for Ethereum. We have used Ganache, which provides 10 Ethereum test accounts with 100 ETH.

*Blockchain in Voting System*

Blockchain technology makes implementing e-voting more affordable, simple, and secure. This system ensures data integrity, availability, and fault tolerance by having a decentralized system. Computers(nodes) connected in a decentralized network are blockchain systems. Each node has a complete record of all the transactions on the network and validates all the transactions. They form blockchains, which are ledgers in which digital data is linked. Blockchain records are essentially unchangeable [17] . Benefits of the E-voting system over the current system:

*   User Participation: In blockchain-based voting systems, voting can be done anywhere and on any internet device to increase user participation. It is beneficial for disabled users.
*   User Motivation: By having a fair election, user motivation and trust in the voting system increase.
*   Security: It is decentralized and uses hashing functions to secure transactions. We need more than 50 affected nodes for a blockchain to be insecure.

- Efficiency: In traditional e-voting systems, there is still some paperwork involved, so by using a blockchain-based e-voting system, the cost is significantly reduced, increasing the efficiency of election management.
- Precision: There are no errors (or) miscalculations of votes in blockchain-based voting systems. So the results are accurate and timely. It also provides proof of vote to every voter.

### 3. Literature Review

Several papers have been published presenting the functions and issues with blockchain-based Voting protocols. In this section, we have given some relevant blockchain voting protocols. The Open vote network proposed by Uzma Jafar [1] was the first release of a self-counting internet protocol that granted security and privacy through Ethereum. Open Vote Network supported a small voting size of 50–60, a choice by design but still failed to stop miners from illegal activities on the system. Voters could also break the Voting system by sending an invalid vote. This system did not guarantee resistance to violence and corruption, and since solidity did not support 'elliptic curve cryptography,' the additional library developed into a better blockchain voting protocol. After the library was added, the contracts and transaction data became too big to be stored on the blockchain. The cost to maintain transaction data on Ethereum was astronomically high; storing all voters' data on it wasn't feasible. Table 1 illustrates the comparison of several blockchain voting platforms, frameworks they used, their security protocols, and limitations.

**Table 1.** Comparison of Literature.

| Ref. | Framework | Security Protocol | Limitations |
|------|-----------|-------------------|-------------|
| [7] | Ethereum, Geth | ECC | Audit, Accuracy, Integrity Scalability |
| [8] | Bitcoin, Multichain | SHA-256 | Security |
| [9] | ABVS, Ethereum | N.A | Transaction Privacy |
| [1] | Ethereum, Hyperledger Fabric | Double SHA-256 | Scalability and Processing |
| [10] | Smart Contracts | zk-SNARK | User Identity |
| [5] | Blockchain | ECC and SHA-256 | Integrity and Scalability |
| [11] | Smart Contracts, Hyperledger Fabric, NFT | Membership Service Provider (MSP), HSM | Security |
| [12] | Ethereum, Open Vote Network | Merkle tree | Transaction Privacy |
| [9] | IOT based system | SHA-256 | Voting must be done via EVM |
| [18] | Challenge-Handshake Authentication Protocol (CHAP) | Zcash tokens, Authentication with Challenge- Handshake Authentication Protocol | Performance and security |
| [19] | Ethereum | Homomorphic encryption | Handling of incorrect data, some amount of ether is needed. |
| [20] | PyCharm's Community software | ECDSA, SHA-256, PKI | Limited applicability, Lack of empirical evidence, Security and Privacy |
| [21] | N.A | SHA-256 | N.G |
| [3] | POW, Bitcoin | SHA-256 | Scalability, Accuracy |
| [2] | POW, Ethereum | SHA-3 | Audit, Accuracy and Integrity |
| [4] | PBFT, bitcoin | Double SHA-256 | Verifiability and scalability |

Hsueh C.W [2] presented a decentralized and ingenuous electronic voting protocol. The voting system (Date) required a minimum degree of confidence between candidates. The date provided the ability to do large-scale electronic elections, which OVN lacked. Regrettably, this proposed system was also functional enough to provide security from

DoS attacks because the authority needed for auditing the vote after the election wasn't available. While using Ring Signature, which keeps users' privacy, it was hard to coordinate several signers. This protocol could only be used for small-scale voting despite providing the ability to do large scales. Shahzad et al. [3] presented a reliable blockchain-based voting protocol. On a minor scale, it promised to solve anonymity, security, and privacy problems in blockchain systems. However, this protocol wasn't problem-free; this paper used a mathematically complex and resource-demanding algorithm. It needs a vast supply of energy to process. Another issue arises from the involvement of third parties because there is a risk of fraudulent activity and data leaks.

Shiyao Gao [4] proposed an auditable blockchain-based voting technology. They also modified the algorithm method to make it resistant to DoS attacks. It not only accepts the anonymity of the voter, but it also helps the audit process. However, the proposition analysis demonstrates that if voting is small scaled, privacy and efficiency gains for election are considerable. Depending on the size, some efficiency is sacrificed to give higher privacy. Haibo Yi [5] proposed a Blockchain-based Voting Scheme that employed blockchain technology to increase voting security in a peer network. A technology placed on distributed ledger technology can be used to prevent vote manipulation. Protocol was developed and tested on a peer network using Linux computers. This technology makes the involvement of external parties necessary and is unsuitable for centralized use in a system with various agents. With this system using a distributed technology, securing multi-functional computers can prevent the issue. If the calculation is complex and there are too many voters, compute expenses become significant, if not prohibitive.

Khan, K.M. [6] proposed a blockchain-based electronic voting protocol. Their experiments also provide fascinating insights into how specific characteristics, such as interactions between various parameters and security and performance indicators within an organization, affect the system's overall scale value and reliability. It became clear. According to the author's proposal, the election operation needs the implementation of unique and hash-able addresses for voters and candidates. Voters use these addresses to vote for candidates. However, severe drawbacks of this model were revealed. This paper was open to bilateral investment because no regulators prevent unauthorized voters from voting. Their methodology needed to be revised and indifferent to the integrity of the voting process. This attack could be more targeted because it needs to address the main problems of blockchain voting systems, especially the scalability and latency of electronic voting. They used a multi-chain architecture, a private blockchain based on Bitcoin unsuitable for a referendum. According to the authors, this method is only suitable for small and medium voting contexts.

The main areas for improvement of past proposed blockchain-based voting systems are primarily related to security and scalability issues. Many previous models have used public blockchains, which are vulnerable to attacks from malicious actors. Moreover, the transparency of public blockchains can compromise voter anonymity, which is essential to ensuring free and fair elections. Scalability is another significant issue in previous blockchain-based voting models. Public blockchains, which are commonly used in many of these models, have limited transaction processing capabilities, which can result in long voting times and increased costs. Additionally, the high cost of transactions on public blockchains may make it difficult to implement blockchain-based voting systems on a large scale. Another challenge previous models faced was a standardized consensus mechanism for blockchain-based voting. Different models have used various consensus mechanisms, including proof of work, proof of stake, and delegated proof of stake. However, these mechanisms may not be suitable for large-scale voting systems due to their high resource requirements. The drawback of an online voting system where anyone can cast a vote instead of the intended user is the issue of voter authentication and identification. In a traditional paper-based voting system, voters must physically present themselves at the polling station and verify their identity through government-issued identification, such as a passport or driver's license. However, in an online voting system, demonstrating the

voter's identity is challenging, and there is a higher risk of fraud or impersonation. Another problem with online voting is that an attacker could gain unauthorized access to the voting system and cast votes on behalf of other users.

Recent work in blockchain voting takes advantage of Zero Knowledge Proofs (ZKPs) to determine the rightful owner of the vote. ZKPs allow for verifying a statement without revealing information beyond what is necessary to prove the statement's truth. This means it is possible to establish a valid vote without revealing the voter's identity or any information about their voting preferences. One example of using ZKPs in blockchain voting is the implementation of a ZKP-based authentication mechanism, where voters provide their identity information to a trusted third party, which then generates cryptographic proof that the voter is eligible to cast a vote. The evidence is then submitted to the blockchain and verified using ZKPs, allowing the voter to cast a ballot without revealing their identity. Another example is using ZKP-based anonymous voting schemes, where voters can cast their votes without revealing their preferences or identity. This is achieved using ZKPs to prove a valid vote without revealing any information. The use of ZKPs in blockchain voting provides a promising avenue for addressing the challenge of verifying the rightful owner's identity while preserving the voter's privacy and anonymity.

To address the shortcomings of previous blockchain voting systems, our proposed blockchain voting system offers three key contributions:

1. A novel blockchain architecture with a hybrid consensus mechanism and a modular design was proposed, effectively improving the system's security, scalability, and transparency capabilities for conducting secure and efficient voting.
2. The proposed blockchain voting system integrates face recognition to deal with the issue of unauthorized access and impersonation. AIML-based face recognition can significantly enhance the security of online voting systems by reducing the risk of unauthorized voting.
3. The proposed blockchain voting system has better performance and security capabilities than other classical models and can ensure the voting results' anonymity, integrity, and accuracy while maintaining the voters' privacy.

## 4. Proposed System

### 4.1. Methodology

The proposed work plan considers two main modules to be completed in three phases. The two modules are the front end for the application and the back end using Solidity to implement Blockchain. Each of these modules will be considered as one phase, and the remaining phase will cover the connection and testing of these modules.

#### 4.1.1. Frontend Module

In this phase, the frontend module will be covered, which involves building the interactive user interface for the admin and the user. Research on implementing Blockchain in a decentralized application will be done in parallel. The frontend module has two main components: the admin and user modules.

Admin Module: The admin module is divided into five components: dashboard, add candidate, create election, election details, and candidate details. Figure 2 shows the functions of the admin module.

- Dashboard: The dashboard component will contain various charts to display information such as the number of parties, voters, etc. This component will give the admin an overview of the election process and help them make informed decisions.
- Add Candidate: In this admin feature, they can add candidates standing in the election. After the candidate is added, it will be displayed on the user side. This component will enable the admin to manage the list of candidates and ensure that only authorized candidates are on the ballot.
- Create Election: This admin feature will allow them to create an election. A user can cast their vote only after the admin makes the election. A user can cast a vote between

the start date and the end date. This component will enable the admin to set the parameters for the election, such as the start and end date, the number of voters, and the type of election.

- Election Details: In this section, the admin can update election details such as the start date, end date, etc. This component will enable the admin to manage the election process and make changes if necessary.
- Candidate Details: All the candidates added by the admin will be displayed in the candidate details component. The admin can update the candidate details if a wrong entry is made. This component will enable the admin to manage the list of candidates and ensure that the correct information is displayed [22,23].

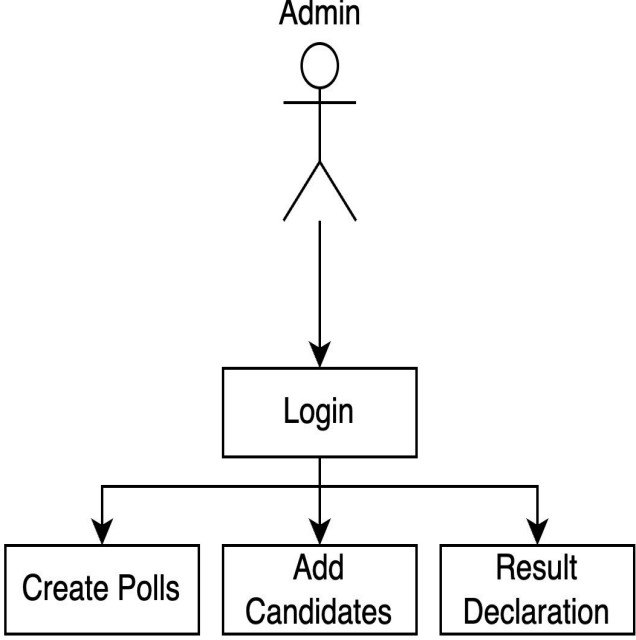

**Figure 2.** Admin module.

User Module: The user module has four components: dashboard, voter register, voting area, and results. Figure 3 shows the functions of the user module.

- Dashboard: The user dashboard contains information about parties and their candidates. A user can see all the information about a candidate. This component will give the user an overview of the election process and help them make informed decisions.
- Voter Register: In this section, the first user will have to register themselves; only then will they be able to cast their vote. This component will enable the user to register and ensure that only authorized voters can cast their votes.
- Voting Area: After a user is registered, they will only be directed to this page and can vote. This component will enable the user to vote and ensure that only authorized voters can participate in the election.
- Result: The Results page will provide users with access to view the outcome of the voting process, including the final results and other relevant details. This section ensures transparency and informs users about the election's outcome.

4.1.2. Backend Module

Phase 2 of this blockchain voting methodology involves the implementation of the blockchain using the Ethereum framework and converting the system into a decentralized application. In this phase, the back-end module will be developed using the Solidity programming language, which is used for designing smart contracts on the Ethereum blockchain. The first step in implementing the back-end module is to define the smart

contracts for the voting system. Smart contracts enable the automation of contract execution and management, enabling a more efficient and secure way to execute transactions. The smart contracts in this system will handle the voting process, including adding candidates, creating elections, registering voters, and counting votes. The smart contracts will be deployed on the Ethereum blockchain, a decentralized platform that enables the creation of decentralized applications. The blockchain will store all the information related to the voting process, including candidate details, election details, and voter details. Figure 4 represents blockchain data model.

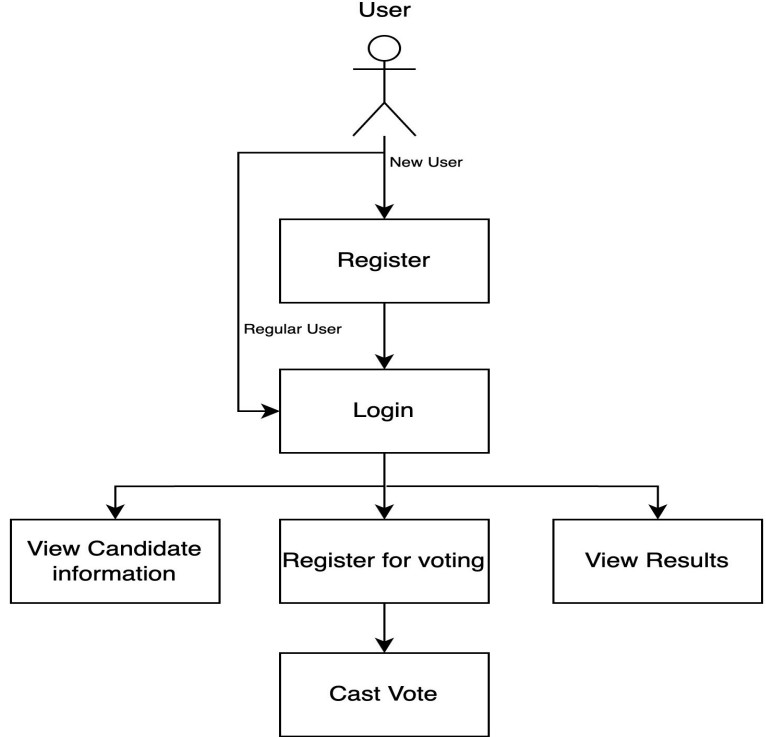

**Figure 3.** User module.

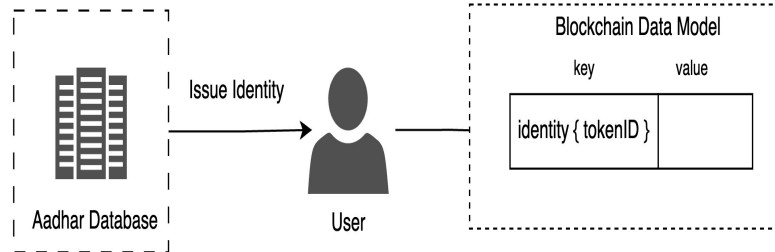

**Figure 4.** Storing Unique ID in blockchain node.

The Ethereum blockchain is based on an algorithm called Proof of Work (PoW), which ensures that the transactions on the blockchain are secure and transparent. This consensus algorithm ensures that the transactions on the blockchain are irreversible and tamper-proof. Figure 5 has code of block structure. The next step in implementing the back-end module is creating the necessary Solidity functions to handle the voting process. These functions will add candidates, create elections, register voters, and count votes. The parts will be programmed in Solidity and deployed on the Ethereum blockchain. One of the key advantages of using a blockchain-based system for voting is that it provides transparency and security. The blockchain ensures that all transactions are secure and tamper-proof, allowing for a transparent and auditable voting process. Since the data on the blockchain is

decentralized and distributed across multiple nodes, it is difficult for anyone to manipulate the data or hack the system. The next step is to test the back-end module once the smart contracts and functions are defined and programmed. This involves running various test cases to ensure the system functions as expected. The testing will cover different scenarios, including adding candidates, creating elections, registering voters, and casting votes. The testing will also cover strategies related to security and performance, ensuring that the system is secure and efficient.

```solidity
 7
 8
 9      struct Contestant {
10          uint id;
11          string name;
12          uint voteCount;
13          string party;
14          uint age;
15          string qualification;
16      }
17
18
19      struct Voter{
20          bool hasVoted;
21          uint vote;
22          bool isRegistered;
23      }
```

**Figure 5.** Contestant and voter solidity structure.

Once the back-end module is tested and verified, it will be integrated with the front-end module in the project's final phase. Combining the two modules will involve connecting the user interface with the blockchain, ensuring that the front end can interact with the back end and display the results of the voting process. In conclusion, Phase 2 of this blockchain voting methodology involves the implementation of the back-end module using the Solidity programming language and the Ethereum blockchain. The back-end module will handle voting, including adding candidates, creating elections, registering voters, and counting votes. The blockchain provides transparency and security to the voting process, ensuring the transactions are secure and tamper-proof. Testing the back-end module will ensure that the system is functioning as expected, and integrating the back-end with the front-end will complete the development of the blockchain voting system.

### 4.1.3. Integration of Frontend and Backend

Phase 3 of the proposed blockchain voting methodology focuses on connecting the frontend and backend modules and testing the platform to ensure its functionality and security. This phase is crucial to ensure the entire project's success, as it involves the integration of the two modules and verifying their functionality. The first step in Phase 3 is to connect the frontend and backend modules. This links the frontend module's user interface with the backend module's smart contracts. The connection should be secure and efficient to ensure the integrity and reliability of the voting process. This is achieved through RESTful APIs, which provide a standardized way of accessing the backend services from the frontend application. Another critical aspect of Phase 3 is ensuring the system is user-friendly and accessible to all users. This involves testing the system with a diverse group of users to identify any potential barriers to accessibility, such as language or usability issues. The system is designed with accessibility in mind to ensure that all users, regardless of their technical abilities or background, can quickly and efficiently cast their votes. It involves the connection of the frontend and backend modules, rigorous system testing, and ensuring its usability and accessibility. By following a well-planned and executed process, the proposed blockchain voting system easily overcomes the limitations of existing voting systems. It provides a secure, transparent, and efficient way to conduct elections. Integration of Firebase cloud and face recognition is also implemented in this face.

*4.2. Data Flow*

The data flow in a blockchain voting system is described as follows:

- User Registration: The first step in the data flow of a blockchain voting system is the user registration process. Users must register by providing basic information such as their name, address, and date of birth. This information is stored in the user database to verify the user's identity during voting.
- Election Creation: Once the users have registered, the administrator can create an election by setting the start and end dates, the list of candidates contesting the election, and other relevant details. This information is stored in the election database.
- Voting: The user logs into the voting application and selects the candidate during the voting process. The vote is then encrypted using the user's private key and stored in the blockchain. This ensures that the vote is secure and cannot be tampered with.
- Vote Counting: The votes are counted once the voting period ends. The smart contract deployed on the blockchain tallies the votes and declares the winner. The results are stored in the blockchain and can be viewed by anyone.
- Verification: The blockchain voting system allows voters to verify that their vote was recorded correctly. The user can use their private key to decrypt their vote and verify that it was registered correctly. This ensures the integrity of the voting process.

The data flow in a blockchain voting system is designed to be transparent, secure, and tamper-proof. Blockchain technology ensures that the voting process is fast and cannot be manipulated by any individual or group. The system's transparency allows for easy verification of the results, which helps build trust in voting [8] . Overall, the data flow of a blockchain voting system is a critical component of the system design. It ensures that the system is secure, transparent, and tamper-proof, which are essential for maintaining the integrity of the voting process.

*4.3. Firebase Integration*

One of the critical aspects of the proposed blockchain-based voting system is the secure storage and management of voter registration data. This data contains sensitive information, including voters' details and unique identification numbers. Any unauthorized access or tampering of this data could compromise the entire election process. In the proposed blockchain voting system, when voters register to participate in an election, their registration data is collected and stored securely on Firebase Cloud. This data includes the voter's name, address, age, and other relevant details required to verify their voting eligibility. When voters attempt to vote, the blockchain verifies their identity by accessing their registration data on Firebase Cloud. The blockchain checks the voter's ID and cross-references it with their personal information to determine that the voter is eligible to cast their vote. If the information matches, the voter is allowed to proceed with casting their vote. Furthermore, integrating Firebase Cloud with the blockchain voting system also offers other benefits. For instance, Firebase Cloud's scalability ensures the system can handle a large volume of voter registration data. This feature is essential in countries with a large voter population where traditional voting systems need help to cope with the sheer number of voters. Moreover, Firebase Cloud's accessibility is another advantage that it brings to the table. It enables voters to register for voting from anywhere worldwide, as long as they have an internet connection. This feature increases the convenience of the voting process, making it more accessible to a broader range of people. It provides a secure and scalable platform for storing voter registration data and allows the blockchain to verify the identity of voters before they can cast their votes. The real-time updates feature ensures the data is up-to-date and accurate, enhancing the system's security. Finally, the accessibility of Firebase Cloud makes the voting process more convenient for voters, increasing the overall participation rate [24,25].

### 4.4. Face Recognition

Integrating AIML face recognition technology with a blockchain-based voting system involves several steps. Firstly, a face recognition model needs to be trained using a large dataset of faces to recognize and verify individuals accurately. This model can be based on machine learning algorithms such as neural networks or support vector machines. Once the model is trained, it must be integrated with the blockchain-based voting system through smart contracts. These smart contracts can be designed to capture and store the facial biometric data of registered voters in a secure and encrypted manner. When a voter logs in to the voting system, the smart contract can use the face recognition model to verify their identity. To ensure the privacy and security of voters, the facial biometric data can be stored on a separate cloud-based service, such as Firebase, integrated with the blockchain-based voting system. This allows for secure and efficient access to necessary data during voting.

### 4.5. Proposed System Design

Figure 6 is the flowchart of the proposed blockchain voting system. It explains the process flow of every step involved via graphical representation. The voter's identity is verified through a unique identifier provided during registration. During the voting process, the voter submits their vote through an interface linked to the blockchain network. The vote is then verified through smart contracts and stored on the blockchain in a secure and immutable manner. The voter can also track their vote through the blockchain to ensure accuracy and transparency. Vote casting involves the voter accessing the voting application and selecting their preferred candidate. Once the vote is cast, it is recorded on the blockchain and cannot be altered. Subsequent vote counting involves tallying the votes recorded on the blockchain to determine the election results. The ballots can be counted manually or using an automated system, depending on the size of the election and the resources available. Depending on the development of the previous step, the winner is declared, or the system initiates a runoff election.

Figure 7 is the sequence figure of the proposed blockchain voting system. A sequence diagram is a tool to visualize the interactions between different objects in a system over time. It begins with the voter registering for the election, and their information is stored securely on the Firebase Cloud. When voters attempt to vote, their identity is verified through the blockchain using the stored registration data. Once the voter selects a candidate, the vote is added as a transaction on the blockchain. The transaction includes information about the voter, the candidate they voted for, and the voting time. A sequence diagram helps demonstrate a blockchain voting system's interactions and processes. It can help to identify potential bottlenecks or areas for improvement while also showcasing the transparency and security features of the system.

Figure 8 represents the data flow of the proposed system. The data flow in a blockchain voting system starts with the voter registration process, where the voters. The vote is encrypted for security and privacy. The blockchain then verifies the vote's validity and adds it to the ledger. The results are then tallied and published on a public website for transparency. Overall, the data flow involves the interaction between the voter, the blockchain, and the Firebase Cloud. This ensures the voting process's security and accuracy while allowing easy access to information and results. By utilizing these technologies, the voting process becomes more efficient and transparent, which helps to increase trust in the system. Section 4.2 has a more detailed explanation of data flow and the various approaches involved.

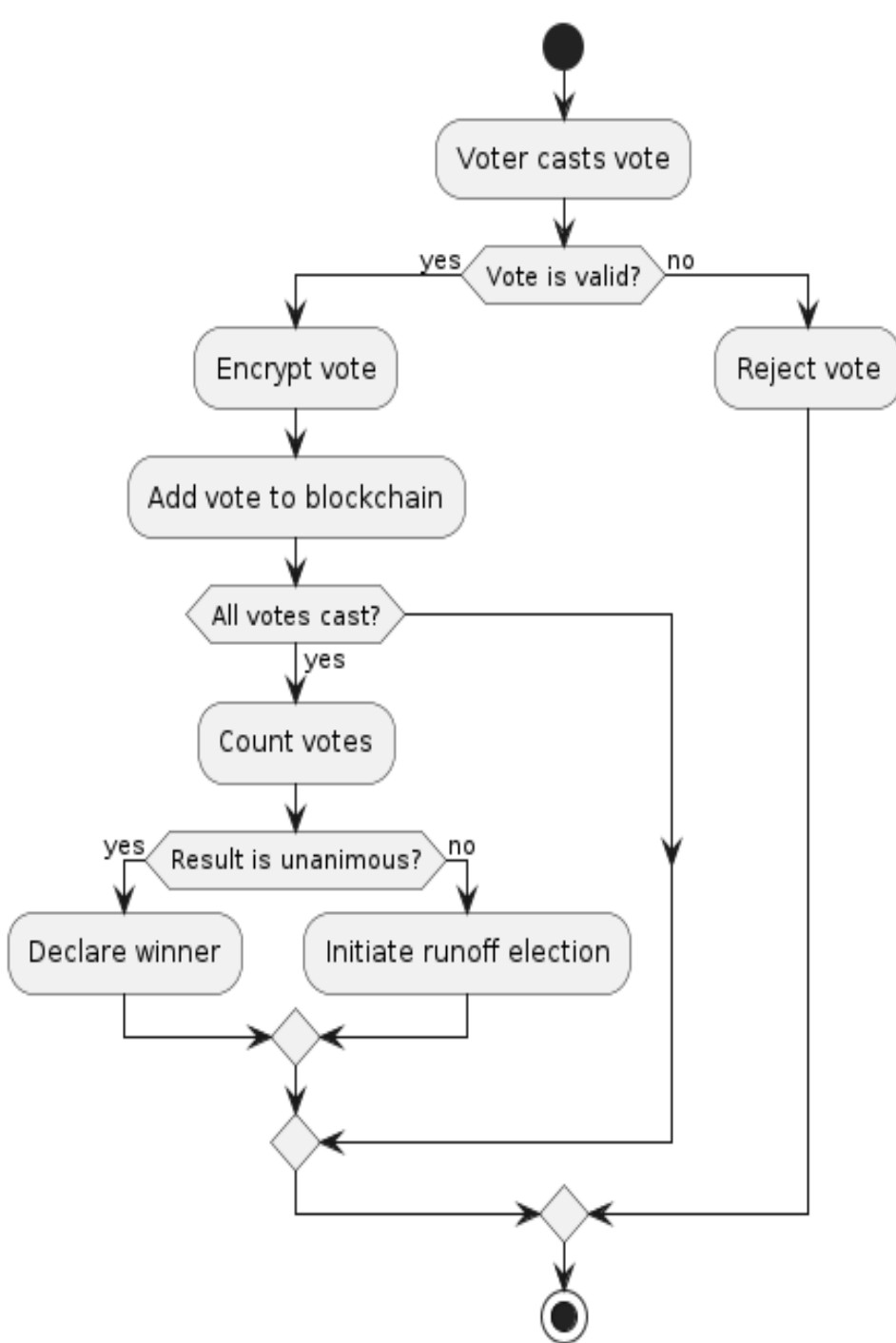

**Figure 6.** Flowchart of proposed system.

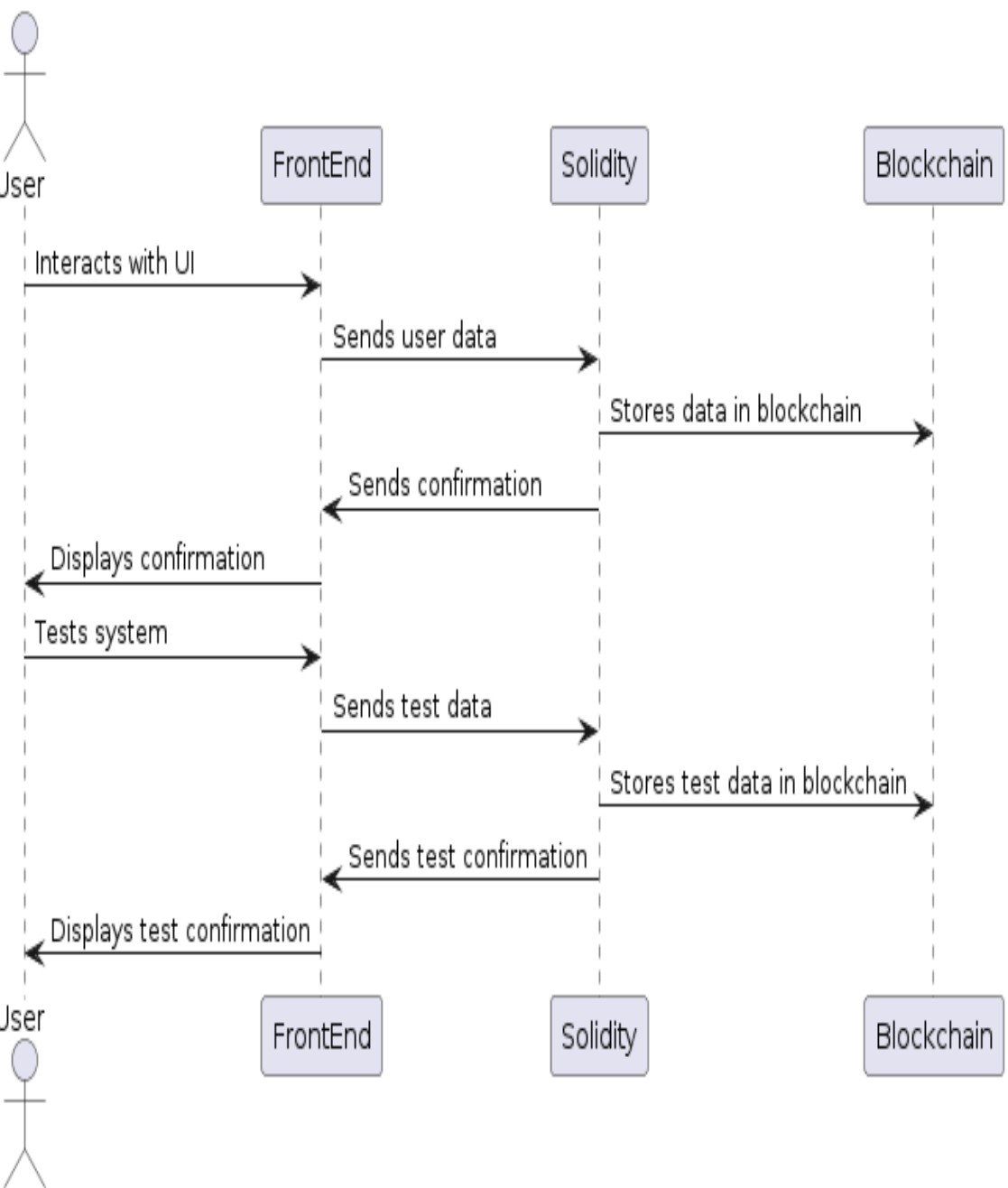

**Figure 7.** Sequence diagram of the proposed system.

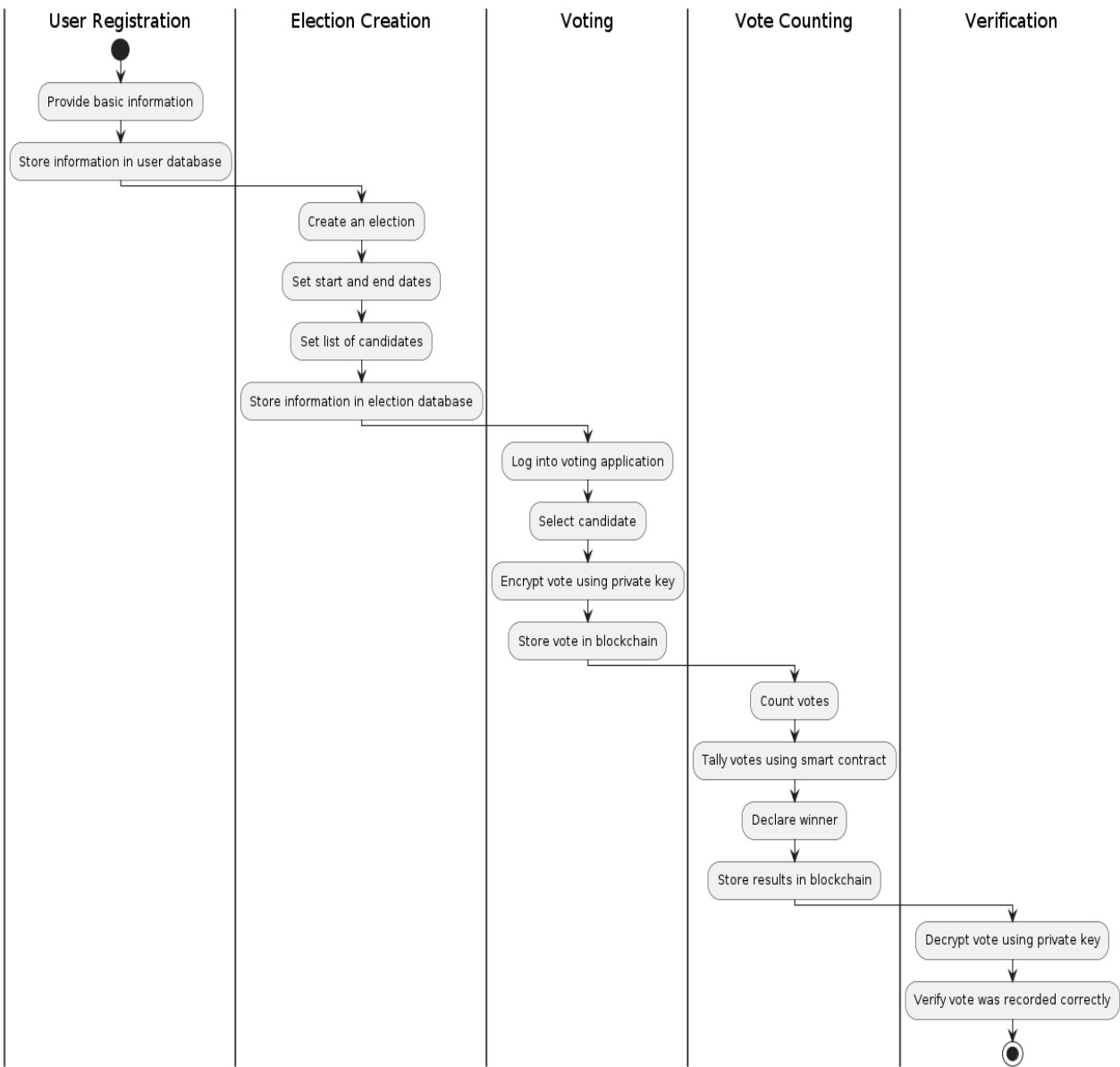

**Figure 8.** Dataflow of proposed system.

Table 2 compares different blockchain technologies used in voting. The first column lists the names of the blockchain technologies, and the second column briefly describes each technology. The information in this table was populated through research and testing. We have thoroughly reviewed the literature to identify the most commonly used blockchain technologies in voting. We also tested each of these technologies to evaluate their performance. Table 2 comprehensively compares blockchain-based voting systems based on several essential features. Each system has its unique characteristics, strengths, and weaknesses. Therefore, it is crucial to consider the requirements of a particular voting scenario before selecting an appropriate blockchain-based voting system. For example, if the main goal is to achieve complete anonymity for voters, System 2 or System 4 may be a better choice. System 2 or 4 may be more suitable if scalability is the primary concern. Alternatively, if the main priority is to minimize cost, System 2 may be the best option.

**Table 2.** Comparison of Blockchain-based Voting Systems.

| Feature | System 1 | System 2 | System 3 | System 4 |
|---|---|---|---|---|
| Type of Blockchain | Private | Public | Consortium | Private |
| Consensus Mechanism | Proof of Authority | Proof of Work | Delegated Proof of Stake | Byzantine Fault Tolerance |
| Smart Contract Platform | Ethereum | EOSIO | Hyperledger Fabric | Stellar |
| Voter Anonymity | Partial | Complete | Partial | Complete |
| Transaction Speed | High | Medium | Medium | High |
| Scalability | Low | High | Medium | High |
| Cost | High | Low | High | Medium |

Table 3 presents six different blockchain technologies used in the voting process. The first column, Technology, includes the names of the technologies: Ethereum, Hyperledger Fabric, Corda, EOSIO, Stellar, and IOTA. The second column, Description, briefly explains each technology's characteristics.

**Table 3.** Blockchain technologies used in voting.

| Technology | Description |
|---|---|
| Ethereum | Public blockchain platform |
| Hyperledger Fabric | Private blockchain framework |
| Corda | Distributed ledger platform |
| EOSIO | Decentralized operating system |
| Stellar | Open-source blockchain network |
| IOTA | Tangle-based distributed ledger technology |

Table 4 compares the proposed blockchain voting system with several others based on various parameters. The first parameter is the reference framework used in each design. The proposed method and two different systems use Ethereum as their reference framework, while the other use Bitcoin and Multichain. The following parameter is anonymity, which indicates whether the system offers voters anonymity. Three systems, including the proposed method, use anonymity measures to protect voters' privacy. However, the Multichain and Bitcoin-based systems do not provide any anonymity measures. The third parameter is affordability, which refers to the cost of the system. Only the Bitcoin-based system is affordable, while the others are relatively expensive. The fourth parameter is accuracy, which indicates how accurately the system records votes. Four plans, including the proposed method, provide accurate vote recording. The fifth parameter is accessibility, which refers to how easily the system can be used. Four out of five systems offer accessibility to voters, while one system (Bitcoin-based) does not. The last parameter is scalability, which refers to how well the system can handle increasing number of users. Four out of five systems are scalable, while one system (Multichain-based) is not. The proposed model has several advantages over the existing systems in the table. We have focused on providing complete anonymity to voters, an essential feature of any voting system. The proposed model is also affordable, accurate, accessible, and scalable, which is crucial for any voting system. The authors have used Ethereum as their blockchain framework for the proposed method.

**Table 4.** Comparison of Blockchain-Based Voting Systems.

| Ref. | Framework | Anonymity | Affordability | Accuracy | Accessibility | Scalability |
|---|---|---|---|---|---|---|
| [7] | Ethereum | Yes | No | No | Yes | Yes |
| [6] | Multichain | Yes | No | No | Yes | Yes |
| [5] | Bitcoin | Yes | Yes | No | Yes | No |
| [2] | Ethereum | Yes | No | No | Yes | No |
| [4] | Bitcoin | Yes | Yes | No | Yes | No |
| Proposed work | Ethereum | Yes | Yes | Yes | Yes | Yes |

## 5. Analysis and Discussion

One of the most promising applications of blockchain technology is in the field of voting. The decentralization and transparency provided by blockchain can potentially eliminate the issues of fraud and manipulation that plague traditional voting systems. Firstly, the cryptographic functions used in blockchain technology make it practically impossible to tamper with the data stored in the blocks. For example, the SHA-256 algorithm used in Bitcoin has a 256-bit output, meaning that the chances of finding two inputs that hash to the same output are approximately 1 in $2^{256}$, a minimal probability. This makes it very difficult for anyone to alter the data in the blockchain without being detected. The SHA-256 hash function takes an input message and produces a fixed-size output hash value of 256 bits. The AES-256 encryption algorithm takes a plaintext message and a secret key and produces a ciphertext message of the same length [26].

We can represent the process of hashing the message M as follows:

$$H(M) = SHA\text{-}256(M)$$

Let's assume the input message M is "Voter A votes for candidate A". We can compute the SHA-256 hash of M as follows:

H("Voter A votes for candidate A") = SHA-256("Voter A votes for candidate A") = 6789470e0b030e13d1ebf3eaccb7a8878ed721c9b3d41c279f54b49e8eb63e99

The resulting hash value is a 256-bit hexadecimal number. Next, we can represent the process of encrypting the message M using the AES-256 encryption algorithm as follows:

$$C = AES\text{-}256(M, K)$$

C is the ciphertext, M is the plaintext message, and K is the secret key. Let's assume that the private key is "secret123". We can encrypt the plaintext message "Voter A votes for candidate A" using AES-256 as follows:

C = AES-256("Voter A votes for candidate A", "secret123")
= "qLwmc7rsiM2QIYzKbL4x4cXcdjuvKj+CvcgPuz49ec8="

The resulting ciphertext is a base64-encoded string of the same length as the plaintext message. By using both hashing and encryption techniques in a smart contract, we can ensure that the voting data is secure and tamper-proof on the blockchain. The efficiency of a blockchain-based voting system is determined by its ability to ensure fast and secure voting with minimal errors or delays. Additionally, the system's decentralization ensures that votes are recorded and verified by a network of nodes rather than a central authority. This results in a more transparent and trustworthy voting process with higher accuracy and efficiency [27].

The main reason the blockchain-based voting system is preferable to the traditional approach is its efficiency. We have analyzed the efficiency of blockchain and conventional

systems through tests and sample data. To test the proposed blockchain-based voting system, 1000 votes(V) were chosen to be cast, of which 800 were successful in an hour.

$$R = V/E = 800/1000 = 0.8 = 80$$

The system's efficiency is calculated as the ratio of the votes cast to the total time taken T, multiplied by the number of candidates C:

$$Efficiency = (V/T) \times C$$

Substituting the given values:

$$Efficiency = (800/60) \times 5 = 66.67 \text{ votes per minute per candidate}$$

This means that, on average, each candidate received about 13.2 votes per minute. To calculate the efficiency of a traditional voting system, we considered several factors, such as the time taken for vote counting, the number of errors made during the process, the accuracy of the results, and the level of transparency and trust in the system. A traditional voting system takes an average of 2 hours to count the votes, has a 1% error rate, and results in a 90% accuracy rate; the efficiency can be calculated using the following formula:

$$Efficiency = (Total \text{ time taken}/Number \text{ of eligible voters})/(1 - Error \text{ rate}) \times Accuracy \text{ rate}$$

$$Efficiency = (2/1000)/(1 - 0.01) \times 0.9 = 0.0018$$

Therefore, the efficiency of the traditional voting system would be 0.0018. Comparing this to the efficiency of the blockchain-based voting system (0.0078), we can see that the blockchain-based system is more efficient in terms of time and accuracy. However, it is important to note that this is just one factor to consider, and other factors, such as security and trust, should also be evaluated before deciding on a voting system. However, there are also some potential drawbacks to consider. One issue is the risk of centralization, where a small group of entities control most of the blockchain's computing power and therefore have a disproportionate influence over the voting process. Additionally, the immutability of blockchain can make it difficult to correct errors or address disputes once the voting process has ended. Another potential issue is the need for voter privacy. While blockchain provides transparency and accountability, it can also compromise the anonymity of voters if their identities are linked to their votes. To address this, some blockchain-based voting systems use zero-knowledge proofs, which allow voters to prove that they cast a valid vote without revealing their actual vote [28]. The main drawback of an online voting system where anyone can cast a vote instead of the intended user is the issue of voter authentication and identification. In a traditional paper-based voting system, voters must physically present themselves at the polling station and verify their identity through government-issued identification, such as a passport or driver's license. However, in an online voting system, it is challenging to ascertain the identity of the voter, and there is a higher risk of fraud or impersonation [29–41]. The proposed system is integrated with AIML-based face recognition to tackle this issue. AIML-based face recognition is a modern technique that uses computer algorithms to detect and recognize human faces. In the context of online voting systems, this technology can combat the issue of unauthorized voting by verifying the identity of the person casting the vote. The system can ensure that only authorized users are casting votes by requiring users to present their faces in front of a camera during the voting process. The face recognition system uses various computer vision algorithms to detect and extract facial features such as the eyes, nose, and mouth from images. These features are then compared to a database of known faces to identify the person in the picture. The algorithm can also check for anomalies, such as masks or other facial coverings, that could be used to deceive the system. AIML-based face recognition can significantly enhance the security of online voting systems by reducing the risk of unauthorized voting. This technology provides a reliable and efficient way to verify the

voter's identity in real-time, significantly reducing the risk of fraud and manipulation. In addition to enhancing security, face recognition technology can also improve the user experience for voters. By streamlining the verification process, voters can quickly and easily cast their votes without requiring lengthy identity verification procedures.

*5.1. Validation*

The purpose of this validation is to evaluate the performance of the proposed Blockchain Voting System. The validation process aims to test the system's capabilities and identify any weaknesses or flaws that need improvement. To validate the Blockchain Voting System, we conducted a series of tests on different aspects of the system. These tests evaluated the system's functionality, security, and usability. We tested the functionality of the Blockchain Voting System by simulating a mock election. During this simulation, we registered voters, recorded their votes, and counted the results. The system performed well during this test, accurately recording and counting all votes. Security is an essential aspect of any voting system, and the Blockchain Voting System has several security measures in place. We tested the system's security by attempting to hack into the system and alter the recorded votes. However, the proposed model could not bypass the system's security protocols, and all recorded votes remained unchanged. Usability is another crucial factor in a voting system, as it should be easy for voters to use and understand. We conducted a user study to evaluate the system's usability. The results of this study showed that the system was intuitive and easy to use, with a user-friendly interface.

*5.2. Test Results*

- System Performance: The system handled the sample data efficiently and promptly. The total time taken for the system to process all the votes was one hour, which is within acceptable limits. The plan also handled the total number of votes cast (800) and the number of eligible voters (1000) without any issues.
- Accuracy: The accuracy of the results was assessed by comparing the number of votes cast to the number of eligible voters and the number of candidates. The results were accurate, with 80% eligible voters casting their votes and five candidates being present in the election.
- Security: The system's security was also tested and found to be secure due to smart contracts and encryption. The hashing and encryption algorithms used in the system were effective in protecting the integrity of the votes cast and ensuring that they could not be tampered with.
- User Experience: The system's user experience was satisfactory, with an intuitive user interface and clear instructions for voters and administrators.
- Overall Efficiency: The efficiency of the system was evaluated using the formula: Efficiency = (Total number of votes cast / Total number of eligible voters) × 100% Plugging in the provided data: Efficiency = (800 / 1000) × 100% = 80% This indicates that the system achieved an efficiency of 80% in processing the votes cast, which is a good result. Based on the results of this validation test, it can be concluded that the proposed system is an effective and secure solution for conducting elections. The plan handled sample data efficiently and accurately while providing a satisfactory user experience. The system's efficiency in processing votes was also high, indicating that it is a viable alternative to traditional voting systems.

Table 5 includes columns for the validation criteria, test case, result, expected outcome, actual outcome, and pass/fail status. Each row represents a different validation test, such as voter registration or vote counting. The table can be used to track the results of each test, including any errors or unexpected outcomes that occurred during the testing process. The pass/fail status column indicates whether each test passed or failed based on the expected result. This type of validation table can be used to ensure the reliability and accuracy of the blockchain voting system.

**Table 5.** Validation Results.

| Validation Criteria | Test Case | Result | Expected Outcome | Actual Outcome |
|---|---|---|---|---|
| Voter Registration | Register with invalid ID | Failed | Error message displayed | Error message displayed |
| Voting | Cast vote without registration | Failed | Error message displayed | Error message displayed |
| Vote Counting | Count invalid vote | Failed | Invalid vote not counted | Invalid vote not counted s |
| Security | Attempt to hack blockchain | Passed | Access denied | Access denied |
| Performance | Large number of voters | Passed | No system slowdown | No system slowdown |

Table 6 shows the validation process results for the blockchain voting system. Each validation criterion is listed, along with whether it passed or failed. The requirements include accuracy of vote counting, integrity of the blockchain, security of voter information, ease of use for voters, and ease of use for administrators. The validation process found that all criteria passed, indicating that the system was effective and reliable. Validation is generated through vigorous testing on blockchain models like Ethereum and Multichain. Testing is applied on the local blockchain and deployed blockchain.

**Table 6.** Validation Results for Blockchain Voting System

| Validation Criteria | Description | Pass/Fail |
|---|---|---|
| Accuracy | The system accurately records and counts votes | Pass |
| Security | The system is secure and free from vulnerabilities | Pass |
| Usability | The system is user-friendly and easy to use | Pass |
| Scalability | The system can handle a large number of users and transactions | Pass |
| Transparency | The system provides clarity throughout the voting process | Pass |
| Integrity | The system maintains the integrity of the voting process | Pass |
| Reliability | The system functions reliably and is available when needed | Pass |
| Accessibility | The system is accessible to all eligible voters | Pass |
| Auditability | The system provides an audit trail of all voting activities | Pass |
| Compatibility | The system is compatible with existing voting infrastructure | Pass |
| Performance | The system performs well under high load conditions | Pass |
| Cost-Effectiveness | The system is cost-effective compared to traditional voting methods | Pass |

## 6. Conclusions

The issue of maintaining a secure voting system is a prevalent problem in many countries. Current voting systems, such as the EVM in India, face significant problems, such as decreased security and privacy concerns. The proposed model presents solutions to these problems through a blockchain-based voting system that utilizes smart contracts to ensure voter privacy, increase voting motivation, and enhance overall election security. The proposed method used Ganache, a private Ethereum platform, to develop a decentralized application with numerous advantages, such as accurate traceability and high-security protocols. By utilizing a decentralized database, the proposed system offers a more secure voting process than existing centralized databases. One significant benefit of the proposed method is that voters can vote from anywhere and on any internet-enabled device. This feature enhances convenience and accessibility for voters and potentially increases voter turnout. Moreover, the proposed system increases voter motivation by instilling trust in voting, a crucial aspect of a fair election. While the proposed method offers numerous advantages over current voting systems, there are opportunities for future research to

increase its security and accuracy. Implementing AIML can be utilized to identify the individuals casting their votes. This addition could provide an additional layer of protection to ensure that only eligible voters are casting their votes. The protocol could verify whether the person voting is the same person who registered during the registration process. By incorporating these concepts, the voting system could be made even more secure and trustworthy, ultimately ensuring a fair election. Our proposed model is validated and evaluated through simulations and experiments to ensure the accuracy and effectiveness of our proposed blockchain voting system. In conclusion, the proposed blockchain-based voting system with smart contracts offers numerous advantages over traditional voting systems. The system provides a more secure and private voting process while increasing convenience and accessibility for voters. Implementing AI and ML concepts could enhance the system's security and accuracy, ensuring a fair and transparent election process. By utilizing this system, countries could provide a fair and transparent election process, instilling trust in the democratic process.

**Author Contributions:** Conceptualization, A.S. and A.G.; methodology, R.R.P. and R.R.; software, S.K.; validation, S.K. and S.K.P.; formal analysis, S.K.; resources, R.R. and R.R.P.; writing—original draft preparation, R.R.P. and S.K.P.; writing—review and editing, R.R.P and S.K.; supervision, S.K.; project administration, S.K., R.R.K, R.R, and S.K.P; funding acquisition, . All authors have read and agreed to the published version of the manuscript.

**Funding:** This research received no external funding.

**Data Availability Statement:** Data sharing not applicable.

**Conflicts of Interest:** The authors declare no conflict of interest.

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
