# Peer review of "Secure Voting Website Using Ethereum and Smart Contracts"

_asi, doi:10.3390/asi6040070_

Round 1

Reviewer 1 Report

The paper presents a blockchain-based voting system that utilizes smart contracts to ensure voter privacy, increase motivation to vote, and enhance overall election security. The topic is appealing. However, the following comments need to be addressed to improve the work:

1. The main drawback of the current work is the validation. The authors shall validate their proposed system. Otherwise, the research has no value.

2.  Section 2. block chain should be a single word with no spaces.

3. The citation format shall be fixed. For example. Hsueh C.W [2], Haibo Yi [5], Khan, K.M. [6], etc.

4. The study does not provide a detailed analysis of the results. What has been presented is only the proposed model!

5.  A detailed comparison of the current work with the previous model should be made to demonstrate the benefit of the proposed model.

6. All tables are not mentioned in the text. The authors should explain each table.

7. The discussion shall be written and presented appropriately.

8.  The authors need to explain explicitly the implications of the research

9.  English must be improved throughout the manuscript.

English must be improved throughout the manuscript.

Reviewer 2 Report

This paper presents a Secure Voting website using Ethereum and smart contracts. In general, the topic is timely and interesting. The manuscript might be improved by addressing the following issues.

1. The biggest concern is that the contribution of this manuscript is unclear. It's suggested to emphasize that in the paper.

2. Many contexts are common in the literature and could cite the source and reduce the corresponding description in order to focus on the core contribution.

3. It is unclear whether the type of paper is a survey paper or a research article. 

1. The usage of blockchain is not consistent. Some places use "Block chain". 2. The manuscript needs to be carefully checked. There are a lot of typos and errors, even in the abstract. 

Reviewer 3 Report

The authors present a design for a D-app that is used for e-voting supported by a Blockchain network. The authors provide some design details. There are some points that need to be clarified:

- It is not clear if the application has being implemented and tested in all the blockchain networks that are mentioned. For example, it is not clear how Table 2 is populated.

- The process described has been covered lately in recent works (especially the last 2 years) but the main problem is that it is not difficult to count a vote, but it is difficult to prove that the person that casts the vote is the rightful owner of this. For example, the father votes for his/her child. This is not addressed at all at the described solution.

- Recent works in the field take also advantage of Zero Knowledge Proofs to address the problem above. Their use should be justified and included in any work that aims to provide a solution that could be of use.

- The number of references is fine but a higher number of recent works would help the authors enhance their own.

The English are ok. One thing to consider is that the authors write very long text. for example on Section 3 (Literature Review) there is only 1 paragraph. This is not helping the reader to follow their arguments.

Round 2

Reviewer 1 Report

The paper has been improved. However, the citations format needs to be checked and fixed. In addition, some errors and typos need to be corrected. For instance,

- This paper proposed a blockchain-based voting system that makes voting easier, simpler, secure, and efficient. 1Lastly;

- A sequence diagram is a tool used to visualize the interactions between different objects in a system over time. t begins with the voter registering for the election, and their information is  stored securely on the Firebase Cloud.

English should be improved 

Author Response

Thanks for the comments. All typos and citation formatting issues are resolved now. English is improved using professional Grammarly software. 

Reviewer 2 Report

Revision has big improvement. No other comments.

Revision has big improvement. No other comments.

Author Response

(The authors gave the same response as above.)
